# Identification of Mir-182-3p/FLI-1 Axis as a Key Signaling in Immune Response in Cervical Cancer: A Comprehensive Bioinformatic Analysis

**DOI:** 10.3390/ijms24076032

**Published:** 2023-03-23

**Authors:** Eric Genaro Salmerón-Bárcenas, Miguel Angel Mendoza-Catalan, Ángela Uray Ramírez-Bautista, Rafael Acxel Lozano-Santos, Francisco Israel Torres-Rojas, Pedro Antonio Ávila-López, Ana Elvira Zacapala-Gómez

**Affiliations:** 1Departamento de Biomedicina Molecular, Centro de Investigación y de Estudios Avanzados del Instituto Politécnico Nacional, CINVESTAV, Ciudad de México 07360, Mexico; 2Laboratorio de Biomedicina Molecular, Facultad de Ciencias Químico Biológicas, Universidad Autónoma de Guerrero, Chilpancingo 39070, Mexico; 3Department of Biochemistry and Molecular Genetics, Feinberg School of Medicine, Northwestern University, Chicago, IL 60611, USA

**Keywords:** miR-182-3p, FLI-1, methylation, AP2α, response immune, cervical cancer

## Abstract

miRNAs modulate gene expression and play critical functions as oncomiRs or tumor suppressors. The miR-182-3p is important in chemoresistance and cancer progression in breast, lung, osteosarcoma, and ovarian cancer. However, the role of miR-182-3p in cervical cancer (CC) has not been elucidated. Aim: To analyze the role of miR-182-3p in CC through a comprehensive bioinformatic analysis. Methods: Gene Expression Omnibus (GEO) databases were used for the expression analysis. The mRNA targets of miR-182-3p were identified using miRDB, TargetScanHuman, and miRPathDB. The prediction of island CpG was performed using the MethPrimer program. The transcription factor binding sites in the FLI-1 promoter were identified using ConSite+, Alibaba2, and ALGGEN-PROMO. The protein–protein interaction (PPI) analysis was performed in STRING 11.5. Results: miR-182-3p was significantly overexpressed in CC patients and has potential as a diagnostic. We identified 330 targets of miR-182-3p including FLI-1, which downregulates its expression in CC. Additionally, the aberrant methylation of the FLI-1 promoter and Ap2a transcription factor could be involved in downregulating FLI1 expression. Finally, we found that FLI-1 is a possible key gene in the immune response in CC. Conclusions: The miR-182-3p/FLI-1 axis plays a critical role in immune response in CC.

## 1. Introduction

Cervical cancer (CC) is the most commonly diagnosed female cancer worldwide, with 4280 deaths reported and 14,100 new cases estimated in 2022. CC ranks in the top ten female cancers in incidence and mortality in California and Texas [1]. Various studies have shown that the alteration in gene expression is an important event in CC [2,3].

miRNAs can regulate gene expression and are a class of non-coding single-stranded small RNA of about 23 nucleotides in length that bind to the 3′ untranslated region (3′UTR) of their specific target mRNAs [4]. Even though miRNAs are coded only by about 3% of human genes, they can regulate about 30% of human protein-coding genes. miRNAs have a wide range of targets and can regulate various biological functions in cancer including metabolism, growth, immunity, etc. [5].

Some studies have investigated the role of miR-182-3p in several types of cancer including osteosarcoma [6], lung [7], oral [8], breast [9,10], and ovarian cancer [11]. In breast cancer, miR-182-3p was found to decrease TRF2 abundance at the telomere, regulating its structure and function and promoting telomeric and pericentromeric DNA damage [9]. In addition, the circulating levels of miR-182-3p were increased in the sera and peripheral blood mononuclear cells with high predictive value [10]. The above suggests miR-182-3p as a biomarker and therapeutic target [9,10].

The role of miR-182-3p in CC is not known. Therefore, this study aimed to analyze the role of miR-182-3p in CC through a comprehensive bioinformatic analysis.

## 2. Results

### 2.1. Mir-182-3p Expression Increases in Cervical Cancer and Could Regulate the FLI-1 Expression

We analyzed the miR-182-3p expression in CC samples compared with the normal tissue samples from the GSE30656 and GSE86100 datasets. We found an overexpression of miR-182-3p in CC (Figure 1A,B). The expression level of miR-182-3p can be used to diagnose CC; it presented a sensitivity of 89% and a specificity of 70% (Figure 1C). In addition, the prognostic value of miR-182-3p expression was analyzed in CC samples from the TCGA dataset using Kaplan–Meier curves. However, the high expression of miR-182-3p did not correlate with the OS (Figure 1D).

To elucidate the role of miR-182-3p in CC, we searched for the miR-1882-3p target in 3′UTR mRNA using the miRDB, TargetScanHuman, and miRPathDB databases. We found 330 target mRNAs in common in these databases (Figure 1E). GO analysis was performed using the DAVID database to understand the biological processes of these 330 target mRNAs of miR-182-3p; the top ten biological processes are shown in Figure 1F. Several key processes in CC were identified such as angiogenesis and cell migration. In addition, we also observed cell differentiation at the top. FLI-1 was selected for future analysis because its role in CC remains unknown. As shown in Figure 1G, miR-182-3p interacts with the 3′UTR of FLI-1 mRNA. We performed a correlation analysis between the seven miR-182-3p and FLI-1 expression in CC samples from the TCGA dataset using the StarBase v2.0 database. Surprisingly, miR-182-3p and FLI-1 expression negatively correlated (R2: −0.215) (Figure 1H). Overall, these results suggest that miR-182-3p could be involved in the decrease in FLI-1 expression in CC.

### 2.2. FLI-1 Expression Decreases in Cervical Cancer

To investigate the FLI-1 expression in CC, we analyzed its expression in CC from the TCGA dataset using the GEPIA database. We found that FLI-1 expression was decreased in CC samples compared with the normal tissue samples (Figure 2A). To confirm these results, we analyzed the FLI-1 expression in the GSE7803 dataset using the GEO database, and similar results were obtained (Figure 2B). In addition, we analyzed the FLI-1 expression in precancerous lesions and CC samples compared with normal tissue samples from the GSE63514 dataset. We found that FLI-1 expression decreased according to CC progression (Figure 2C). To evaluate the FLI-1 expression in response to chemotherapy, we analyzed its expression in the CC samples of patients with chemotherapy resistance (CR) and compared it with the CC samples of patients with no chemotherapy resistance (NCR) to chemotherapy from the GSE56363 dataset. We observed that FLI-1 expression was decreased in the CC patient samples with CR (Figure 2D). Finally, the prognostic value of FLI-1 expression in CC was analyzed through Kaplan–Meier curves in the TCGA dataset using the Kaplan–Meier Plotter database. The results revealed that low expression of FLI-1 is associated with RFS (Figure 2F), but not with OS (Figure 2E). Altogether, these results suggest that FLI-1 expression is decreased in CC and could be helpful as a prognostic biomarker.

### 2.3. Methylation in FLI-1 Promoter Increases in Cervical Cancer

To determine the DNA methylation in the FLI-1 promoter in CC, we searched a CpG island in the FLI-1 promoter using Methprimer program. As shown in Figure 3A, we found a CpG island located in −1500 to +1500 around TSS. Then, we analyzed the methylation level in the FLI-1 promoter in the CC samples compared with the normal tissue samples from the TCGA dataset using the DiseaseMeth database. The results revealed that the methylation in the FLI-1 promoter was increased in CC (Figure 3B). To confirm this result, we analyzed the methylation in the FLI-1 promoter in two independent datasets (GSE30760 and GSE46306) using the GEO database, and similar results were obtained (Figure 3C–F). Finally, we performed a correlation analysis between the expression and methylation of FLI-1 in the CC samples from the TCGA dataset using the cBioPortal database. The results showed a negative correlation between the expression and methylation of FLI-1 (Spearman: −0.56, Pearson: −0.54 and R2: 0.29) (Figure 3G). These results suggest that the hypermethylation at the FLI-1 promoter could promote the low expression of FLI-1.

### 2.4. AP2α Expression Increases in Cervical Cancer and Could Downregulate FLI-1 Expression

To identify the transcription factors (TFs) involved in the downregulation of FLI-1 in CC, we performed a prediction of TFBSs in the FLI-1 promoter using the CONSITE and ALIBABA programs. As shown in Figure 4A, only two TFs in common were identified including SP1 and AP2α. To confirm these results, we performed a prediction analysis of TFBSs in the FLI-1 promoter with the ALGGEN database using the matrix shown in Figure 4B, and we found TFBSs of AP2α and SP1 in the FLI-1 promoter. Next, we analyzed their expression in the CC samples compared to the normal tissue samples from TCGA using the GEPIA database, and only the AP2α expression was increased in CC (Figure 4C). To confirm this result, we analyzed its expression in the GSE7803 dataset using the GEO database; unfortunately, its expression was not increased in CC (Figure 4D). Therefore, we examined the AP2α expression to protein level by IHC in the CC samples compared with the normal tissue samples from the HPA database. We found that AP2α expression increased in the CC samples (Figure 4E), supporting the results obtained at the mRNA level from the TCGA dataset. Finally, the prognostic value of AP2α expression in the CC samples revealed that the high expression of AP2α correlated with worse RFS (Figure 4G) but not with OS (Figure 4F). These data suggest that AP2α could inhibit FLI-1 expression by binding to the FLI-1 promoter in CC.

### 2.5. FLI-1 as a Possible Key Gene in Immune Response in Cervical Cancer

To explore the pathways and biological processes associated with FLI-1 in CC, we performed a functional enrichment analysis using the Enrich database, taking into account the genes that positively correlated with FLI-1 expression. This correlation analysis was performed in the CC samples from the TCGA dataset using the cBioPortal database; a total of 892 genes that positively correlated with FLI-1 expression were selected. Then, we performed a pathway analysis using the MSigDB Hallmark 2020 library in the Enrich database. The identified pathways are shown in Figure 5A. Several signaling pathways well-known in cancer were identified such as KRAS signaling UP, IL-6/JAK/STAT3 signaling, IL-2/STAT5 signaling, TNF-alpha signaling via NF-kB as well as epithelial–mesenchymal transition and apoptosis.

Interestingly, signaling pathways related to immune response were identified including interferon gamma response, inflammatory response, complement, and interferon alpha response. We performed the same search in the KEGG 2021 Human library in the Enrich database to confirm these results. We present the top ten pathways in Figure 5B. Similarly, we found enrichment pathways related to immune response such as cytokine–cytokine receptor interaction, the chemokine signaling pathway, and chemokine signaling pathway. In addition, GO analysis was performed using the Biological Process 2018 library in the Enrich database, and the top ten biological processes are shown in Figure 5C.

Similarly, biological processes related to immune response were found such as the cytokine-mediated signaling pathway, regulation of the immune response, and inflammatory response (Figure 5C). To validate these results, we analyzed the expression correlation and the regular potential of target genes using the CISTROME database. As shown in Figure 5D, 192 genes were identified in common. Finally, we performed the pathways and biological processes analysis in the MSigDB Hallmark 2020 (Figure 5E), KEGG 2021 Human (Figure 5F), and Biological Process 2018 (Figure 5G) libraries using the Enrich database, respectively. Surprisingly, similar results were found, which validated our previous results. 

Additionally, we selected the top samples with high FLI-1 expression (top 30) and low FLI-1 expression (top 30) from the TCGA dataset (Figure 6A). A total of 4274 DEGs were identified (Figure 6B) that clearly distinguished these two groups (Figure 6C). The GSEA analysis showed an enrichment of pathways related to the immune response in the CC samples with high FLI-1 expression such as allograft rejection, interferon-gamma response, and inflammatory response. At the same time, E2F targets, oxidative phosphorylation, and glycolysis were enriched pathways in the CC samples with low FLI-1 expression (Figure 6D,E). Finally, an immune cell recruitment analysis was performed in CC samples with high FLI-1 expression compared with the CC samples with low FLI-1 expression using the CIBERSORTx tool. Interestingly, the results revealed increased immune cell recruitment in the CC samples with high FLI-1 expression (Figure 7). Altogether, these results suggest that FLI-1 could play a key role in the immune response in CC.

## 3. Discussion

Previous studies have shown that miRNAs participate in cancer progression as oncogenes or tumor suppressors. Today, miRNA represents a tool for next-generation therapy in cancer. miR-182-3p expression and its role in CC have not been elucidated. Therefore, through bioinformatic tools, we analyzed the role of miR-182-3p in CC.

We identified that miR-182-3p is overexpressed in CC and can be used to diagnose CC. In breast cancer, it was shown that the expression of miR-182-3p was significantly increased compared to a healthy individual; the increased onco-microRNA-182-3p inhibited T cells [10]. Additionally, Shi et al. (2020) [12] demonstrated that the proliferation and invasion of nasopharyngeal cancer cells were significantly increased in cells with miR-182-3p overexpression. However, in osteosarcoma (OS), the low expression of miR-182-3p in OS cell lines compared to hMSC was observed. miR-182-3p positively regulated proliferation [6] and the role of miR-182-3p is dependent on the type of cancer.

miR-182-3p interacts with the 3’UTR of FLI-1 mRNA and FLI-1, and miR-182-3p expression exhibits a negative relationship. Our study was the first to demonstrate that miR-182-3p could target FLI-1. The studies reported FLI-1 overexpression in cancer and its role as an oncogene in ovarian cancer [13], melanoma [14], small-cell lung cancer [15], and breast cancer [16]. Gastric cancer demonstrated low-expression of FLI-1 [17]. 

Our results show that FLI-1 down-expression was associated with tumor stage and patient OS. In gastric cancer, Del Portillo et al. (2019) [17] showed low-expression FLI-1. Normal epithelial cells, intestinal metaplasia (IM), and low-grade dysplasia (LGD) showed higher FLI-1 nuclear staining compared to gastric cancer epithelial cells. FLI-1 expression was not associated with tumor stage, but was associated with patient OS. Additionally, FLI-1 suppresses invasion and proliferation in gastric cancer, suggesting that FLI-1 is a tumor suppressor gene and a prognostic biomarker of survival. These results suggest that FLI-1 expression can be a prognostic biomarker of survival.

In addition, we identified factors that could be involved in low FLI-1 expression in CC such as transcription factors (TFs), but our results demonstrate that TFAP2α is overexpressed in CC. TFAP-2α in CC can activate the c-erbB-2 (proto-oncogene) promoter, but it does not stimulate the HPV16 or HPV18 E6/E7 promoter [18]. Additionally, our data demonstrated that AP2α expression presents a high prognostic value for CC. Wu and Zhang (2018) [19] showed in papillary thyroid carcinoma (PTC) that AP2α overexpression was significantly associated with tumor stage, histologic type, and independently predicted shorter OS. Patients with TFAP2α overexpression had a shorter OS than AP2α down-expression, particularly in PTC patients with advanced tumor stages (III and IV).

Finally, we analyzed signaling pathways related to FLI-1 low expression. FLI-1 low expression in CC patients is associated with signaling pathways related to immune response and the number of infiltrating immune cells. On the other hand, Scheiber et al. (2014) [20] demonstrated that overexpression of FLI-1 is related to cytokine–cytokine receptor interaction in breast cancer. In addition, Wang et al. (2020) [21] reported that in breast cancer patients with FLI-1 overexpression, FLI-1 regulates the immune system. FLI-1 high-expression obtained prominent immune cell tumor infiltration compared to patients with FLI-1 low-expression. In addition, B cells, T cells, macrophages, mast cells, monocytes, and NK cells showed significantly fewer cell numbers when FLI-1 expression was decreased.

In summary, our results revealed that miR-182-3p expression was significantly increased in the CC patients and presents an acceptable diagnostic value, and its target mRNA is FLI-1, whose expression was decreased in the samples of patients with CC and correlated with poor relapse-free survival. Moreover, aberrant methylation and AP2α transcription factor could be involved in FLI-1 downregulation in samples of patients with CC. Interestingly, FLI-1 could regulate the expression of genes involved in pathways related to the immune response in patients with CC including cytokine–cytokine receptor interaction, the T-cell receptor signaling pathway, and chemokine signaling pathway. Our results suggest that miR-182-3p could be a key gene in CC and a therapeutic target.

Our study demonstrates that FLI-1 plays an important role in the development of CC. A limitation of this study was the lack of in vitro experiments, but analyses were performed with several databases reporting the same results and using high-confidence intervals. 

## 4. Materials and Methods

### 4.1. Expression Analysis

The expression analysis of miR-182-3p was performed in the CC and normal samples from the GSE30656 and GSE86100 datasets. The expression analysis was performed on the mRNA and protein levels. At the mRNA level, it was performed with samples from TCGA as well as GSE30656, GSE86100, GSE7803, GSE63514, and datasets using the gene expression profiling interactive analysis (GEPIA) [22] and GEO databases [23], respectively. At the protein level, the expression analysis was performed on the CC and normal samples from the Human Protein Atlas (HPA) [24]. The GSE7803 dataset (Platform: GPL96, [HG-U133A]) Affymetrix Human Genome U133A Array) includes 10 normal and 21 CC samples [25]. The GSE63514 dataset (Platform: GPL570, [HG-U133_Plus_2] Affymetrix Human Genome U133_Plus 2.0 Array) includes 24 normal, 14 CIN1, 22 CIN2, 40 CIN3, and 28 CC samples [26]. The GSE56363 dataset (Platform: GPL4133, Agilent-014850 Whole Human Genome Microarray 4 × 44K G4112F (Feature Number version)) includes 12 CC samples of patients with complete response (CR) and nine CC samples of patients with a non-complete response (NCR) to chemotherapy [27]. The GSE30656 dataset (Platform: GPL6955, Agilent-016436 Human miRNA Microarray 1.0 Feature Number version) includes ten normal and 19 CC samples [28]. The GSE86100 dataset (Platform: GPL19730, Agilent-046064 Unrestricted_Human_miRNA_V19.0_Microarray Probe name version) includes six normal and six CC samples [29]. The GSE7803, GSE30656, and GSE86100 datasets were analyzed in the GEO2R program [30]. The differences were calculated using the one-way ANOVA and Student’s t-test for the GEPIA and GEO databases, respectively. A *p*-value of ˂0.05 was considered significant. 

The differentially expressed genes between CC patients with a high and low expression of FLI-1 were identified from the TCGA-CESC dataset [31]. The analysis was performed as previously described [32]. Briefly, differential expression between the high and low FLI-1 groups was performed using TCGA biolinks and DESeq2 packages [33,34]. Transcripts with a fold change ˃1.5 and an adjusted *p*-value < 0.05 were considered differentially expressed genes. In addition, visualization was performed with the ggplot2 package [35]. A *p*-value < 0.05 was considered statistically significant.

### 4.2. Receiver Operating Characteristic (ROC) Curve Analysis

The ROC curve analysis of miR-182-3p expression was carried out for the CC and normal tissue samples from the GSE30656 dataset using easyROC software [36]. Then, the area under the curve (AUC) with a confidence interval (95%), sensitivity, cut-off, specificity, and p-value parameters were calculated.

### 4.3. Survival Analysis

The overall survival (OS) and relapse-free survival (RFS) were analyzed in the CC samples from the TCGA dataset using Kaplan–Meier plotter (KM-Plotter) [37]. The hazard ratio (HR) with 95% confidence intervals (CI) and LogRank *p*-value were calculated considering a *p*-value ˂ 0.05 to be significant.

### 4.4. Identification Analysis of Target mRNAs

The target mRNAs of miR-182-3p were identified and validated using miRDB [38], TargetScanHuman 7.0 [39], and miRPathDBv2.0 [40].

### 4.5. Correlation Analysis

The correlation analysis between the methylation and expression of FLI-1 was performed from the TCGA dataset using the cBioPortal database [41,42]. The correlation coefficients of Spearman, Pearson, and R2 were calculated, and a *p*-value < 0.05 was considered statistically significant. The correlation analysis between the miR-182-3p and FLI-1 expression was performed in the CC samples from the TCGA dataset using the StarBase v2.0 database [43].

### 4.6. Methylation Analysis

The FLI-1 promoter (−2000 to +1000 relative to TSS) was downloaded from The Eukaryotic Promoter Database (EPD) [44] from the Expert Protein Analysis System (ExPASy) portal [45]. The prediction of island CpG was performed using the MethPrimer program [46]. The methylation in the FLI-1 promoter was determined in the CC and normal tissue samples from the TCGA dataset as well as the GSE30760 and GSE46306 datasets using the DiseaseMeth V2.0 [47] and Gene Expression Omnibus (GEO) databases [23], respectively. The TCGA dataset (450k (Illumina Infinium HumanMethylation450 BeadChip)) includes three normal and 307 CC samples [47]. The GSE30760 dataset (Platform: GPL8490, Illumina Human Methylation 27 Bead Chip (HumanMethylation27_270596_v.1.2)) contains 152 normal and 63 CC samples [48]. The GSE46306 dataset (Platform: GPL13534, Illumina Human Methylation 450 Bead Chip (HumanMethylation450_15017482)) includes 20 normal and seven CC samples [49]. The GSE30760 and GSE46306 datasets were analyzed in the GEO2R program [30], the differences were determined using the Student’s t-test, and a *p*-value ˂ 0.05 was considered statistically significant.

### 4.7. Identification Analysis of Transcription Factors Binding Sites

The transcription factors binding sites (TFBSs) in the FLI-1 promoter (−1000 to +100 relative to TSS) were identified using ConSite [50], Alibaba2 [51], and ALGGEN-PROMO [52].

### 4.8. Estimation Analysis of Immune Cell Recruitment

The abundance of immune cell recruitment to the TCGA samples was estimated using CIBERSORTx software [53]. We used the data of patients with high and low expression of FLI-1 as the input. The differences were determined using the Student’s t-test, and a *p*-value < 0.05 was considered significant.

### 4.9. PPI, Pathway Enrichment, and GSEA Analysis

The protein–protein interaction (PPI) analysis was performed in STRING 11.5 [54]. The functional enrichment was analyzed using the Enrich [55] and CISTROME Data Browser databases [56]. The gene set enrichment analysis (GSEA) was performed using GSEA software [57]. The normalized enrichment score (NES) was determined, and a *p*-value < 0.05 was considered statistically significant.

## 5. Conclusions

Through a comprehensive bioinformatic analysis, we found a high miR-182-3p expression in CC and downregulated FLI-1 expression. Additionally, aberrant methylation in the FLI-1 promoter and increased expression of AP2α decreased the FLI-1 expression. FLI-1 is a key TF in the immune response of patients with CC. The miR-182-3p/FLI-1 axis plays an important role in the immune response in CC.

## Figures and Tables

**Figure 1 ijms-24-06032-f001:**
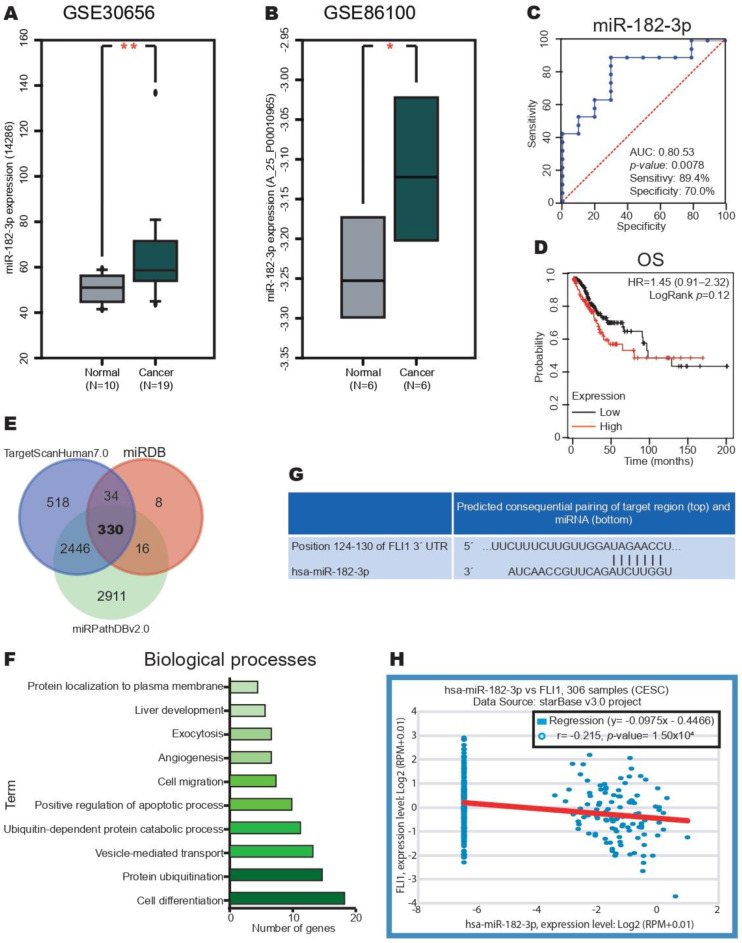
miR-182-3p expression increases in cervical cancer. (**A**) miR-182-3p expression in CC and normal tissue samples from the GSE30656 dataset. (**B**) miR-182-3p expression in CC and normal tissue samples from the GSE86100 dataset. (**C**) Diagnostic value of miR-182-3p. (**D**) Kaplan–Meier curves of OS according to the miR-182-3p expression in CC samples of patients with CC from the TCGA dataset. (**E**) FLI1 mRNA is a potential target in the miR-182-3p, miRTargetLink Human and miRDB databases. (**G**) Pairing between the target region and miRNA from the TargetScanHuman7.0 database. (**F**) Biological processes of these 330 target mRNAs of miR-182-3p. (**H**) Correlation between FLI-1 and miR-182-3p expression in patients with CC from the TCGA dataset. Median cut-off. * *p*-value ˂ 0.05 and ** *p*-value ˂ 0.01.

**Figure 2 ijms-24-06032-f002:**
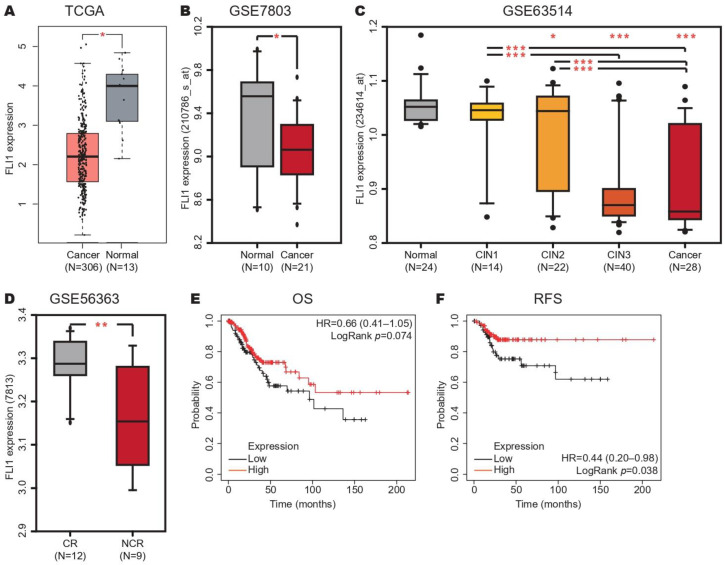
FLI-1 expression increases in cervical cancer. (**A**) FLI-1 expression in the CC and normal tissue samples from the TCGA dataset. (**B**) FLI-1 expression in the CC and normal tissue samples from the GSE7803 dataset. (**C**) FLI-1 expression in the CC, CIN3, CIN2, CIN1, and normal tissue samples from the GSE63514 dataset. (**D**) FLI-1 expression in the CC and normal tissue samples from the GSE56363 dataset. (**E**) Kaplan–Meier curves of OS according to FLI-1 expression in the CC samples of patients with CC from the TCGA dataset. (**F**) Kaplan–Meier curves of RFS according to FLI-1 expression in the CC samples of patients with CC from the TCGA dataset. CIN: Cervical intra-epithelial neoplasia. * *p*-value ˂ 0.05, ** *p*-value ˂ 0.01 and *** *p*-value < 0.001.

**Figure 3 ijms-24-06032-f003:**
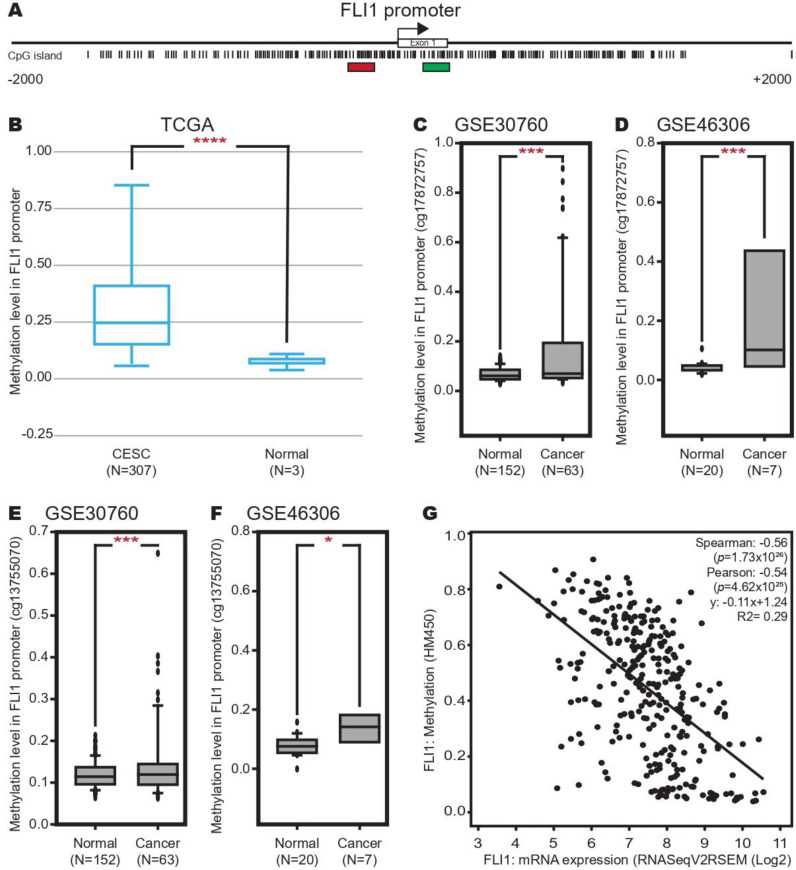
Methylation in the FLI-1 promoter increases in cervical cancer tissue. (**A**) CpG island at the FLI-1 promoter. The CpG island is located in a region ±2 kb around the TSS. Red box: localization of probe cg13755070. Green box: localization of probe cg17872757. (**B**) Methylation level at FLI-1 promoter in the CC and normal tissue samples from the TCGA dataset. (**C**) Methylation level at the FLI-1 promoter in the CC and normal tissue samples from the GSE30760 dataset (probe cg17872757). (**D**) Methylation level at the FLI-1 promoter in the CC and normal tissue samples from the GSE46306 dataset (probe cg17872757). (**E**) Methylation level at the FLI-1 promoter in the CC and normal tissue samples from the GSE30760 dataset (probe cg13755070). (**F**) Methylation level at the FLI-1 promoter in the CC and normal tissue samples from the GSE46306 dataset (probe cg13755070). (**G**) Correlation between the methylation and FLI-1 expression in patients with CC from the TCGA dataset. * *p*-value ˂ 0.05, *** *p*-value < 0.001 and **** *p*-value < 0.0001.

**Figure 4 ijms-24-06032-f004:**
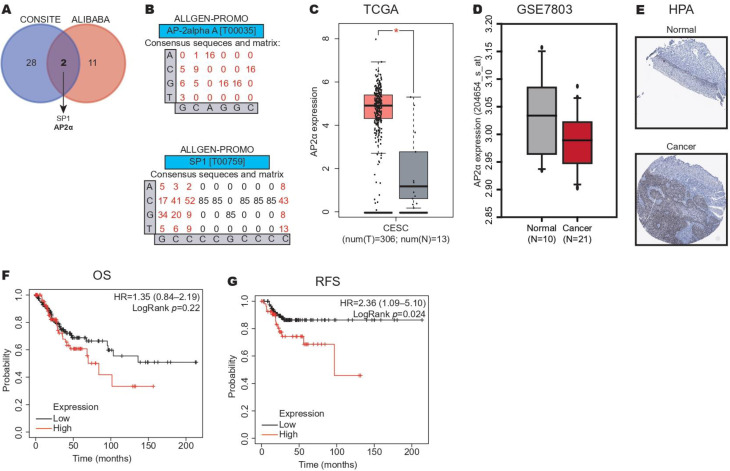
AP2α expression increases in cervical cancer. (**A**) Identification of binding sites for the SP1 and AP2α transcription factors in the FLI1 promoter using the CONSITE and ALIBABA databases. (**B**) Validation of binding sites for SP1 and AP2α transcription factors in the FLI-1 promoter using the ALLGEN-PROMO database. (**C**) AP2α expression in the CC and normal tissue samples from the TCGA dataset. (**D**) AP2α expression in the CC and normal tissue samples from the GSE7803 dataset. (**E**) AP2α expression in the CC and normal tissue samples from the HPA dataset. (**F**) Kaplan–Meier curves of OS according to AP2α expression in the CC samples of patients with CC from the TCGA dataset. (**G**) Kaplan–Meier curves of RFS according to AP2α expression in the CC samples of patients with CC from the TCGA dataset. * *p*-value ˂ 0.05.

**Figure 5 ijms-24-06032-f005:**
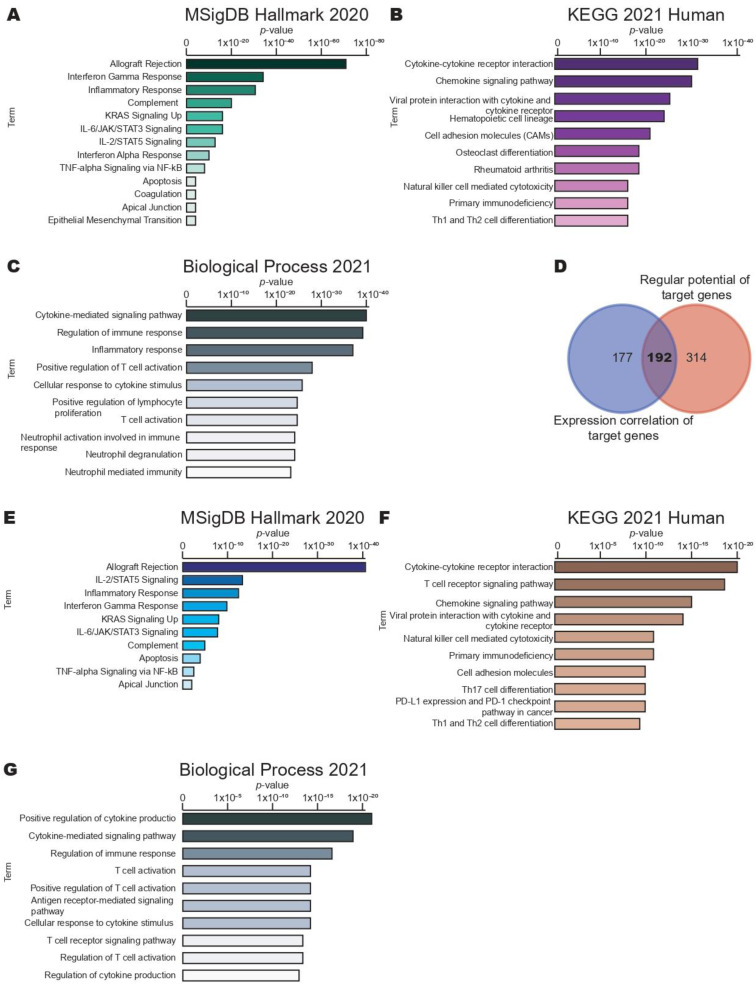
FLI-1 regulates the gene expression participating in the immune response in cervical cancer. (**A**) Identification of pathways according to genes that positively correlate with FLI-1 using the MSigDB Hallmark 2020 library in the Enrich database. (**B**) Validation of identified pathways of genes that positively correlate with FLI-1 using the KEGG 2021 Human library in the Enrich database. (**C**) Identification of the biological processes of genes that positively correlate with FLI-1 using the Biological Process 2021 library in the Enrich database. (**D**) Identification of genes in common considering the regular potential and expression correlation of target genes using the CISTROME database. (**E**) Identification of pathways of genes identified in (**D**) using the MSigDB Hallmark 2020 library in the Enrich database. (**F**) Validation of the identified pathways of genes selected in (**D**) using the KEGG 2021 Human library in the Enrich database. (**G**) Identification of biological processes of genes identified in (**D**) using the Biological Process 2021 library in the Enrich database.

**Figure 6 ijms-24-06032-f006:**
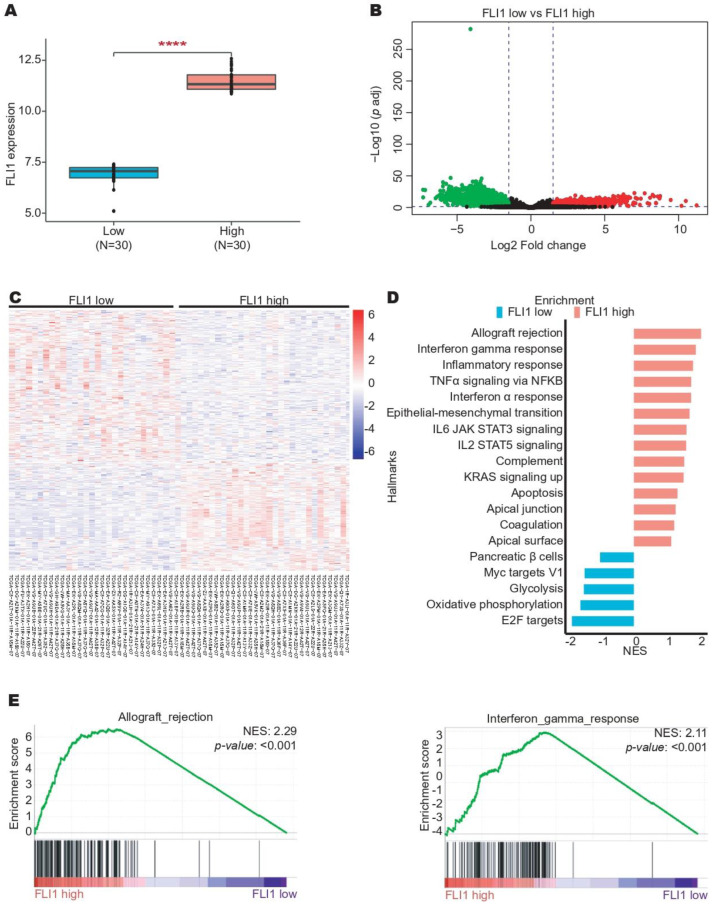
High FLI-1 expression is associated with pathways related to immune response in cervical cancer. (**A**) FLI-1 expression in CC patients with high FLI-1 expression (top 30) and low FLI-1 expression (top 30) from the TCGA dataset. (**B**) DEGs between the CC patients with low FLI-1 and high FLI-1 expression from the TCGA dataset. (**C**) Heatmap with DEGs from the TCGA dataset. (**D**) Identification of pathways enriched considering DEGs using GSEA software. (**E**) Two graphs representative of the GSEA analysis are shown. **** *p*-value < 0.0001.

**Figure 7 ijms-24-06032-f007:**
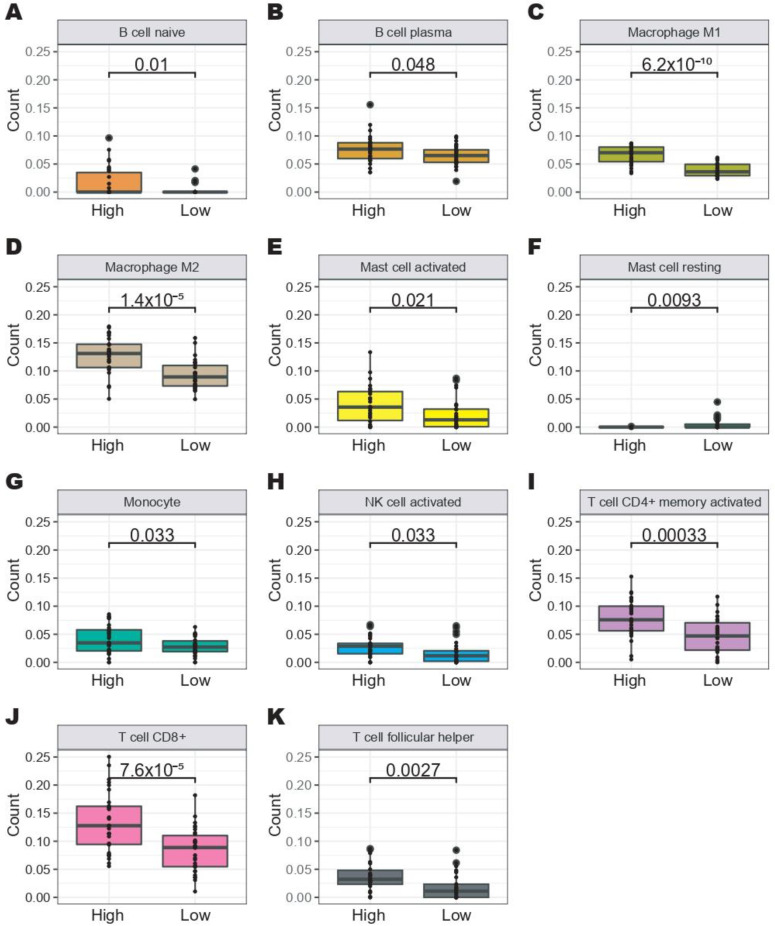
High FLI-1 expression is associated with significant immune cell recruitment in cervical cancer. Quantification of (**A**) B cell naive, (**B**) B cell plasma, (**C**) macrophage M1, (**D**) macrophage M2, (**E**) mast cell activated, (**F**) mast cell resting, (**G**) monocyte, (**H**) NK cell activated, (**I**) T cell CD4+ memory activated, (**J**) T cell CD8+, and (**K**) T cell follicular helper in CC patients with high FLI-1 expression and low FLI-1 expression from the TCGA dataset using CIBERSORTx software. The data used were transformed to log2, which allows the use of the Student *t*-test.

## Data Availability

The raw data used to support the findings of this study will be made available by the authors, without undue reservation, to any qualified researcher.

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
