# Peer review of "Identification of Mir-182-3p/FLI-1 Axis as a Key Signaling in Immune Response in Cervical Cancer: A Comprehensive Bioinformatic Analysis"

_ijms, 2023, doi:10.3390/ijms24076032_

Round 1

Reviewer 1 Report

In the manuscript “Identification of miR-182-3p/FLI-1 axis as key signaling in immune response in cervical cancer: a comprehensive bioinformatic analysis”, the authors explore the expression of miR-182-3p and FLI-1 in normal and cancer tissues. The author found that miR-182-3p expression is significantly increased in CC patients and has potential diagnostic. Further work demonstrated that FLI-1 should be one of the target gene regulated by miR-182-3p. Incomplete manuscript content makes it hard to read. In addition, without any validation, the regulation relationship between miR-182-3p and FLI-1 also not convince enough. There are several comments to help the authors improve their draft. The reviewer will really appreciate if the authors could submit the complete manuscript at the next time.

1. Line 25, could downregulate its expression in CC, this sentence is confusing. What’s “its” represent here, miR-182-3p or its regulated genes?

2. miR-182-3p has been widely investigated in various cancers, such as breast cancer, thyroid cancer, lung cancer, and prostate cancer. A comprehensive summarize of the current research progresses have been made should be added to the introduction section.

3. Line 48-56, this sentence is the main findings and conclusion of this manuscript.  It should be removed or moved to conclusion or discuss section.

4. Where are the figure legends? All the panel from each figure should be clearly described.

5. Where is the figure 1.

6. In Figure 2A, the tumor and normal groups should be labeled clearly in the panel.

Author Response

POINT-BY-POINT RESPONSE

Reviewer 1

In the manuscript “Identification of miR-182-3p/FLI-1 axis as key signaling in immune response in cervical cancer: a comprehensive bioinformatic analysis”, the authors explore the expression of miR-182-3p and FLI-1 in normal and cancer tissues. The author found that miR-182-3p expression is significantly increased in CC patients and has potential diagnostic. Further work demonstrated that FLI-1 should be one of the target gene regulated by miR-182-3p. Incomplete manuscript content makes it hard to read. In addition, without any validation, the regulation relationship between miR-182-3p and FLI-1 also not convince enough. There are several comments to help the authors improve their draft. The reviewer will really appreciate if the authors could submit the complete manuscript at the next time.

We appreciate the reviewer’s comments on our manuscript. Your comments and corrections were important to the improvement of this new version. Thank you

  1. Line 25, “could downregulate its expression in CC”, this sentence is confusing. What’s “its” represent here, miR-182-3p or its regulated genes?

We have realized the correction: “We identified a total of 330 targets of miR-182-3p, including FLI-1, which downregulates its expression in CC”

  1. miR-182-3p has been widely investigated in various cancers, such as breast cancer, thyroid cancer, lung cancer, and prostate cancer. A comprehensive summarize of the current research progresses have been made should be added to the introduction section.

miR-182-3p has not been widely investigated in cancers. Only one study in osteosarcoma, lung, oral, and ovarian cancer have be realized. Breast cancer is reported in two studies. We have summarized our research.

  1. Line 48-56, this sentence is the main findings and conclusion of this manuscript. It should be removed or moved to conclusion or discuss section.

We are considering your accurate comment. We try to add the information in the discussion or conclusion, but it is pertinent to delete the paragraph. Because the summary of the results is mentioned in the paragraph. But, in the discussion, we do not need a summary of all the results in one paragraph because we discuss the results in logical order. In conclusion, the writing is punctual.

  1. Where are the figure legends? All the panel from each figure should be clearly described.

Sorry, we loaded the figure, but it is not displayed in the file. We will be careful. All the panels from each figure are described.

  1. Where is the figure 1.

Sorry, we loaded the figure, but it is not displayed in the file. We will be careful.

  1. In Figure 2A, the tumor and normal groups should be labeled clearly in the panel.

In the figure, we have labeled the tumor and normal groups in the panel.

Note: We have reviewed the manuscript, and the full manuscript has been uploaded. We did not validate our result with in vitro assays of the relationship between miR-182-3p and FLI-1, but the analyzes were performed with several databases, reporting the same results and using high-confidence intervals. This strength for our study. Other working groups can carry out in vitro studies concerning our results; they will cite our study.

Reviewer 2 Report

The authors utilized public sequencing dataset to investigate the carcinogenic effects of miR-182-3p cervical cancer. Different miR softwares were used to find potential mRNA targets and miR-182-3p/FLI1 axis was identified.

Major issues:

1> In Figure7, it is not approrpriate to use t-test to compare proportions of different immune cells between two groups. Because the distribution of the data is not assumed  to be normal distribution or approximate distribution.

2> Rescue experiment can be done to validate the upstream and downstream relationship between FLI1 and miR-182-3p

Minor issues:

1> In materials and method part, there exist some typos.

2> In figure 3A, please maximize the size of CpG island and it is very hard to see it very clearly.

3> In order to keep consistency of labeling of p values, you can replace most p values with asterisks if they are less than cutoff values.

Author Response

Reviewer 2

The authors utilized public sequencing dataset to investigate the carcinogenic effects of miR-182-3p cervical cancer. Different miR softwares were used to find potential mRNA targets and miR-182-3p/FLI1 axis was identified.

We appreciate the reviewer’s comments on our manuscript. Your comments and corrections were important to the improvement of this new version. Thank you

Major issues:

  • In Figure7, it is not approrpriate to use t-test to compare proportions of different immune cells between two groups. Because the distribution of the data is not assumed  to be normal distribution or approximate distribution.

A transformation in log2 was made; this allows to use of the t-student test. Log2 is used when normalizing the expression of genes.

  • Rescue experiment can be done to validate the upstream and downstream relationship between FLI1 and miR-182-3p

We mentioned the limitation of this study, the lack of in vitro experiments is a limitation, but analyses were performed with several databases, reporting the same results and using high-confidence intervals.  The advantage is that only our studies have shown the downstream and upstream regulation of miR-182-3p. This offers the possibility that our results are the basis for future research; other studies in vitro can validate the upstream and downstream relationship between FLI1 and miR-182-3p. They will cite our study.

Minor issues:

  • In materials and method part, there exist some typos.

We have realized the change in materials and method part

  • In figure 3A, please maximize the size of CpG island and it is very hard to see it very clearly.

We increase the size of CpG island. It is easier and clear to see.

  • In order to keep consistency of labeling of p values, you can replace most p values with asterisks if they are less than cutoff values.

We have replaced p values with *, and in the legend of Figure 3B and 6A, the label for the value of p was added. In figure 7, the p-values ​​were not replaced by asterisks because we consider it important to mention the p-value; this could be important for other study groups to select a cell population for to study.

Round 2

Reviewer 1 Report

Thanks for the author’s reply. The authors have already solved parts of my previously comments, and the revised version has been improved to a certain extent. But the draft needs a more professional and careful modification is required before considering publication. There are several comments.

1. The language should be improved remarkably. Some of sentence are difficult to understand. For example, “Identification of miR-182-3p as a potential to target the mRNA using miRTargetLink Human and miRDB databases.”

2. What’s “CIN” represents?

3. normal sample should be “normal tissue sample.

4. In the overall survival analysis, the classification standard to make sure the cutoff value should be provided. Median or optimal cutoff? The n value should be listed in the panel and/or figure legends. In addition, the sources of the data used in the Kaplan-Meier curves should be claimed.

5. In generally, one miRNA can target to multiple target genes.  Why the authors only focused on FLI-1? The logic behind the chose should be claimed.

6. The figures should be re-organized with professional software, for example, Adobe Illustrator. line weight, color, font type, font size, and font color should be unified and keep them looking good.

7. Here, the FLI-1 is regulated in both transcriptional and post-transcriptional modification, but the title does not show it out.

Author Response

POINT-BY-POINT RESPONSE

Reviewer 1

Thanks for the author’s reply. The authors have already solved parts of my previously comments, and the revised version has been improved to a certain extent. But the draft needs a more professional and careful modification is required before considering publication. There are several comments.

  1. The language should be improved remarkably. Some of sentence are difficult to understand. For example, “Identification of miR-182-3p as a potential to target the mRNA using miRTargetLink Human and miRDB databases.”

The manuscript have be revised for better redaction in English, we realized correction in the manuscript.

  1. What’s “CIN” represents?

Thank you for your comments, we have added the full name of “cervical intra-epithelial neoplasia” for NIC in manuscript

  1. “normal sample” should be ““normal tissue sample.”

We have added in manuscript that samples are tissue. 

  1. In the overall survival analysis, the classification standard to make sure the cutoff value should be provided. Median or optimal cutoff? The n value should be listed in the panel and/or figure legends. In addition, the sources of the data used in the Kaplan-Meier curves should be claimed.

The classification standard to make sure the cutoff value was median cutoff

The values ​​of n appear in the figures.

Kaplan-Meier curves were analyzed in CC samples from the TCGA dataset. In methodology is the date.

  1. In generally, one miRNA can target to multiple target genes. Why the authors only focused on FLI-1? The logic behind the chose should be claimed.

We have analyzed several target the miR-182, we performed corrections of expression between miR-182 and yours targets, comparison the expression levels in normal and cancer tissue samples. FLI1 showed to be an important gene in CC.

  1. The figures should be re-organized with professional software, for example, Adobe Illustrator. line weight, color, font type, font size, and font color should be unified and keep them looking good.

We used professional software for realized figure: Adobe Illustrator CC, 64 bits, 2018. But the fugure was send in pdf because it is format requerit in this journal.

  1. Here, the FLI-1 is regulated in both transcriptional and post-transcriptional modification, but the title does not show it out.

Transcriptional and post-transcriptional modification was analyzed because the FlI-1 mRNA is targeted by miR-182. The title of the article includes FLI1 as part of the pathway that can favor the development of cancer.

Reviewer 2 Report

I think it can be accepted in present form.

Author Response

Thank you for your approbation to publish

Round 3

Reviewer 1 Report

Thanks for the author’s reply. The authors have already solved parts of my previously comments, and the revised version has been improved to a certain extent. But the draft needs a more professional and careful modification is required before considering publication. There are several comments.

1. The language should be improved remarkably. Some of sentence are difficult to understand. For example, “Identification of miR-182-3p as a potential to target the mRNA using miRTargetLink Human and miRDB databases.”

2. What’s “CIN” represents?

3. normal sample should be “normal tissue sample.

4. In the overall survival analysis, the classification standard to make sure the cutoff value should be provided. Median or optimal cutoff? The n value should be listed in the panel and/or figure legends. In addition, the sources of the data used in the Kaplan-Meier curves should be claimed.

5. In generally, one miRNA can target to multiple target genes.  Why the authors only focused on FLI-1? The logic behind the chose should be claimed.

6. The figures should be re-organized with professional software, for example, Adobe Illustrator. line weight, color, font type, font size, and font color should be unified and keep them looking good.

7. Here, the FLI-1 is regulated in both transcriptional and post-transcriptional modification, but the title does not show it out.

Author Response

(The authors gave the same response as above.)
